# Low-shot Object Learning with Mutual Exclusivity Bias

**Anh Thai**[1]    **Ahmad Humayun**[2*]   **Stefan Stojanov**[1*]   **Zixuan Huang**[3]
**Bikram Boote**[1]    **James M. Rehg**[1,3]
[1]Georgia Institute of Technology, [2]Google Deepmind,
[3]University of Illinois, Urbana-Champaign

## Abstract

This paper introduces Low-shot Object Learning with Mutual Exclusivity Bias (LSME), the first computational framing of mutual exclusivity bias, a phenomenon commonly observed in infants during word learning. We provide a novel dataset, comprehensive baselines, and a state-of-the-art method to enable the ML community to tackle this challenging learning task. The goal of LSME is to analyze an RGB image of a scene containing multiple objects and correctly associate a previously-unknown object instance with a provided category label. This association is then used to perform low-shot learning to test category generalization. We provide a data generation pipeline for the LSME problem and conduct a thorough analysis of the factors that contribute to its difficulty. Additionally, we evaluate the performance of multiple baselines, including state-of-the-art foundation models. Finally, we present a baseline approach that outperforms state-of-the-art models in terms of low-shot accuracy. Code and data are available at https://github.com/rehg-lab/LSME.

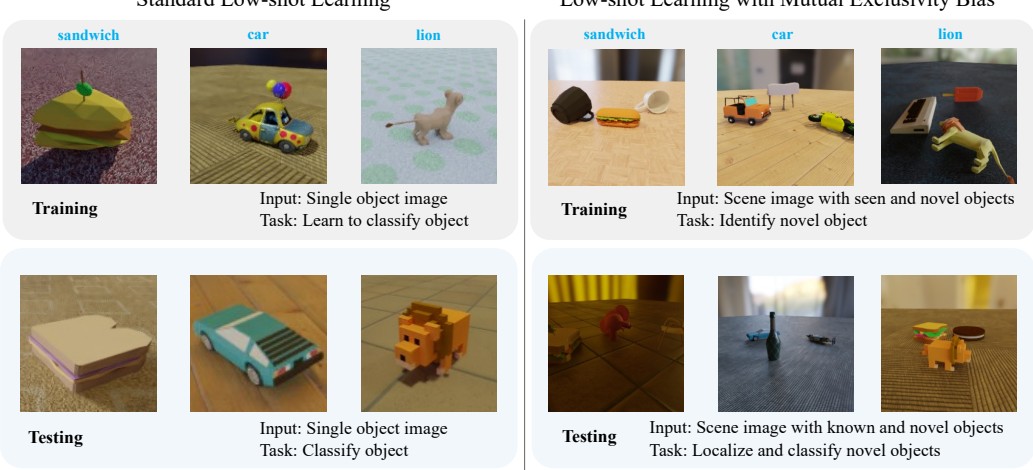

Figure 1: We propose **L**ow-**S**hot Object Learning with **M**utual **E**xclusivity Bias (LSME), a novel setting that is more realistic and significantly challenging than the standard low-shot object learning setting. Given a scene with multiple objects including seen and novel objects, the goal of LSME is to use mutual exclusivity bias to correctly identify the unfamiliar object and generalize this association to other instances of the novel category at testing time.

---

*Equal contribution.

37th Conference on Neural Information Processing Systems (NeurIPS 2023) Track on Datasets and Benchmarks.

# 1 Introduction

Toddlers are incredible learners, acquiring an average of 10 to 20 new words per week during a period known as the vocabulary spurt [29]. This is possible in part because of effective inductive biases that help them to overcome the inherent ambiguity that arises in inferring which real-world object a spoken object name (e.g., provided by a caregiver), is referring to. Inductive biases like shape bias have received more interest [44, 35, 36], due in part to widespread interest in the link between 3D object shape and categorization. In contrast, the key inductive bias of mutual exclusivity has not yet received significant attention [11, 27, 28]. Mutual exclusivity leverages the assumption that each object has only one label. Thus when presented with a scene containing multiple objects, some familiar and some unfamiliar, mutual exclusivity guides the child to bind new object names to the unfamiliar objects. For example, consider a toddler playing with three toys: her favorite Ducky, a Mickey Mouse figure, and a toy that she does not know the name of. When the caregiver says, "Look at the dinosaur," the toddler uses mutual exclusivity bias to rule out Ducky and Mickey, as she already knows their names. She then associates the word "dinosaur" with the unfamiliar object. After this mapping is made, the toddler can retain the category label "dinosaur," and use it later to label other dinosaur-like objects. The goal of this work is to create a novel dataset and associated baselines to spur the ML community in developing computational models for mutual exclusivity, as a further step towards modeling the inductive biases that underlie human object learning performance.

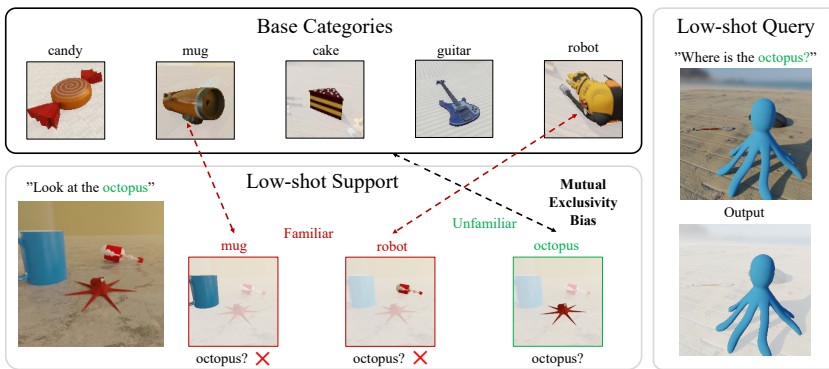

Figure 2: Our proposed LSME (Low-shot Object Learning with Mutual Exclusivity Bias) setting. We leverage mutual exclusivity bias observed in humans to associate a novel word label with an unfamiliar object in a scene. Given a set of familiar categories (base classes), the task is to use this bias to correctly identify the unfamiliar object and generalize this association to other instances of the novel category.

This paper introduces Low-shot Object Learning with Mutual Exclusivity Bias (LSME), a computational framing of mutual exclusivity as an inference problem, with low shot learning as the downstream measure of generalization performance (see Fig. 2 for an overview and Table 1 for related settings). The LSME task begins with a set of already-known object categories. The input is an RGB image containing multiple object instances from known categories and a single object instance of an unknown category, along with a novel category label. Solving the LSME problem requires three steps: 1) Localize the object instances within the scene via segmentation; 2) Associate the novel category label with the unknown category instance that it refers to; and 3) Solve a low-shot learning task by classifying other object instances from the novel category during inference. The ability to reliably solve LSME would be an important step in creating agents that can learn with minimal supervision. For example, household robotics is a long-standing goal of AI. Our work could ultimately yield a robot capable of rapidly learning the names of novel objects in the home, without the need for a burdensome supervised training stage.

LSME can be partitioned into three sub-tasks: 1) object localization, 2) open-world recognition, and 3) low-shot learning. The model must first localize each object in the scene. It must then distinguish between learned and unknown categories to correctly assign the novel label to the unknown instance. Finally, the model must generalize to new instances of the novel categories after only one or a few labeled samples. At the heart of LSME is the challenge of associating novel object instances with their category labels. In supervised pre-training, there is a danger of leakage if the datasets used in pre-training overlap with the low-shot category labels used in testing. In order to prevent this, we

Table 1: Comparative analysis of LSME and other related settings. LSME requires a comprehensive understanding of scenes that involve the presence of multiple objects.

| Settings / Properties | Low-shot Recognition | Open-set Detection | Low-shot Detection | Object Discovery | GCD [45] | NCDL [9] | Open Category Detection | Image Conditioned Detection | LSME |
|---|---|---|---|---|---|---|---|---|---|
| Instance Localization | ✗ | ✓ | ✓ | ✓ | ✗ | ✓ | ✓ | ✓ | ✓ |
| Mutual Exclusivity Bias | ✗ | ✗ | ✗ | ✗ | ✗ | ✗ | ✗ | ✗ | ✓ |
| Discover Novel Classes | ✗ | ✓ | ✗ | ✗ | ✓ | ✓ | ✓ | ✗ | ✓ |
| Label Novel Classes | ✓ | ✗ | ✓ | ✗ | ✓ | ✓ | ✓ | ✗ | ✓ |
| Zero/Low-shot | ✓ | ✗ | ✓ | ✓ | ✗ | ✗ | ✓ | ✓ | ✓ |
| No Pretrained LLM | ✓ | ✓ | ✓ | ✓ | ✓ | ✓ | ✗ | ✓ | ✓ |

constrain the LSME problem design so that pre-training prior to the low-shot phase cannot include any object-label association data (e.g. as in CLIP). This constraint further aligns with the investigation of a key question in developmental psychology: How can a child rapidly learn words in the absence of prior knowledge of object semantics? [51, 26]

It's natural to ask if LSME can be solved by simply combining existing SOTA models. We find that this is not the case. Errors in the first two sub-tasks propagate to the low-shot stage and degrade accuracy. For example, complex spatial interactions (e.g. occlusions) between objects make reliable instance segmentation challenging, leading to poor feature extraction for each object. Furthermore, any errors in identifying novel vs. known objects will result in label noise that degrades low-shot recognition. In addition, pre-trained feature representations from foundation models (e.g. DINOv1 [5] and DINOv2 [33]) are not sufficient to solve our ask effectively (see Sec. 4).

One challenge we address in this paper is the creation of suitable LSME datasets. Since real-world data is not available, we introduce a generic data generation pipeline that can take any categorical 3D dataset as input and render training and testing data with realistic backgrounds, lighting variations, and realisic object poses. Our dataset is structured with increasing levels of complexity in order to aid in understanding the shortcomings of learning approaches.

Additionally, we benchmark the performances of various baselines on LSME, including SOTA foundation models with robust visual representations such as DINOv2 [33], ImageBind [12], and CLIP [37]. Interestingly, we find that the performance of these baselines degrades significantly when there are occlusions/incomplete instance masks (see Sec. 4.3).

From these observations, we propose a baseline that involves pretraining the object representations on cluttered scenes with occlusion. Following the success of prior works that show performance improvement when training on multi-view inputs [43, 17], we propose self-supervised training on the large-scale multi-view multi-object ABC [21] dataset using contrastive learning. Our baseline composed of the robust 2D representations from foundation models and 3D features from multi-view inputs demonstrates improvement in accuracy compared to existing models when occlusions are present by approx. $10\%$. To verify that this gain is significant, we evaluate our approach on the real-world dataset CO3D [38] and show the benefit of multi-view training with occlusion.

In summary, our contributions are 4-fold: 1) The first to provide a computational framing of mutual exclusivity via LSME; 2) A dataset generation pipeline that enables the creation of progressively more challenging LSME tasks using any set of 3D models; 3) Performance benchmarking of multiple baselines including foundation models on our novel task; and 4) A novel baseline method that defines the SOTA on LSME.

## 2 Related Work

Because we are the first to provide a computational framing of mutual exclusivity (ME), there is no direct prior work to compare to. [11] demonstrated that existing deep models fail at ME, but did not provide a comprehensive framing or SOTA methods. We now review three bodies of related work.

**Self-supervised and Weakly-Supervised Learning** Self-supervised learning aims to leverage the inherent structure in visual data to train representations suitable for downstream tasks, without relying on labels. Contrastive methods [7, 15, 8] work by treating image pairs under different data augmentations as positive examples, and other sample pairs as negatives. DINOv1[5] enforces "global-local" correspondences by using multiple image crops, while other methods [48, 55, 31] focus on pixel-level matching or combine both objectives [33]. On the other hand, VISPE [17],

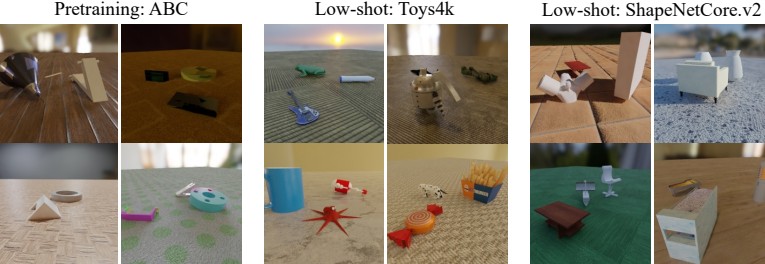

Figure 3: Rendered scenes used for LSME. First column has scenes of ABC [21] objects that are used for pretaining. Second and third columns show data for low-shot evaluation on Toys4k [44] and ShapeNetCore.v2 [6]

DOPE [43], and VSA [23] incorporate multi-view data by treating samples captured from different viewpoints as positive pairs. These approaches leverage 3D structure to gain improvements.

In contrast, recent weakly-supervised learning foundation models such as CLIP [37] and Image-Bind [12] take advantage of multi-modal signals, including language and audio in addition to visual information. Another body of work [14, 52, 50] attempts to learn the underlying visual structure of data by learning to reconstruct images from heavily masked inputs. In this work, we study the performance of these methods on our novel task. Moreover, we extend these representations by incorporating multi-view multi-object information. By integrating data from multiple viewpoints and considering information at the object level instead of the scene or pixel level, we aim to learn object representations that can reason about the spatial layout and category composition of scenes with multiple objects.

**Related Settings** Vaze et al. proposed Generalized Category Discovery (GCD) [45], a setting that allows input to come from both known and unknown distributions, and further aims to label novel data. We differ from this setting in that we do not assume object localization information. We further consider scene-centric data that provides a more holistic view of the entire scene instead of object-centric data as input. This enables our model to take into account the relationships between multiple objects within the scene.

Fomenko et al. [9] introduce the novel class discovery and localization (NCDL) setting, which closely relates to our proposed LSME task. Both tasks share the goal of localizing and classifying unknown objects in a scene. However, we operate in a low-shot setup, where only a limited number of samples from the novel categories are available. This contrasts with NCDL, which focuses on balancing a long-tail distribution of novel classes with varied number of samples per class. In addition, our setting requires mutual exclusivity bias—explicitly learning ambiguous word-object associations.

Open-category object detection and segmentation [53, 19] have recently gained popularity for its generalization ability beyond the trained vocabulary. This task focuses on using large pre-trained vision-language models to detect and segment known and novel objects via prompting the LLMs. In contrast, our task focuses specifically on learning the association between objects and their corresponding labels. These labels may not necessarily align with the vocabulary typically used for LLM pretraining (e.g. using pseudo names for toy objects like "dax" as commonly seen in psychology experiments [42]). LSME can flexibly accommodate various vocabulary systems that potentially is a challenge for pretrained LLMs.

Image-conditioned object detection [30, 34] is a related setting where language input is not required. The goal of this setting is to detect objects that share the same semantic characteristics as the template objects given as one or a few input images. Our setting solves a more general and challenging task where the model is required to identify the unknown object of interest before generalizing to other object instances from the novel category. LSME is further related to low-shot learning [43, 54, 25], open-world learning [46, 4], and object instance segmentation [47, 49, 24] settings as it integrates these components into a holistic solution. (See Tab. 1)

**Computational Frameworks Motivated by Developmental Psychology** Recent works have investigated various learning scenarios to explore the strategies children employ when acquiring new concepts [18, 3, 16]. This work complements these efforts by introducing a comprehensive benchmark that studies the mutual exclusivity bias commonly observed in infants during the initial stages of word learning.

**Synthetic 3D Datasets and Dataset Generators** Synthetic 3D datasets and environments have been explored to study a wide range of tasks [44, 43, 50, 13, 41] where real-world data might not be available at a large scale. Recent realistic data rendering engines [13, 10, 41] and image generation models [39, 40] have significantly reduced the sim2real domain gap, enabling more effective generalization to real-world settings. In this work, we leverage the benefits of synthetic data to tackle a novel problem where real-world datasets are unavailable. This allows us to create diverse and customizable scenarios that closely resemble real-world settings to study the problem in a controlled and scalable manner.

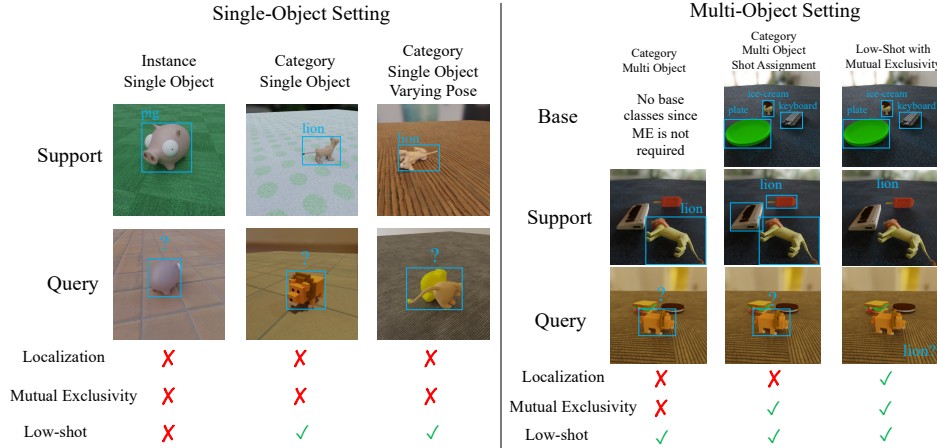

Figure 4: Different variants of LSME setting. **Left:** Single-object case where only a single object is present in the scene for both supports and queries and **Right:** Multi-object setting where multiple objects are present in the scene. Bounding boxes and texts indicate given localization information and labels respectively. The difficulty levels of these variants increase from left to right, with the hardest setting—our proposed LSME on the rightmost column requiring the models to achieve all 3 properties: localization, mutual exclusivity bias, and low-shot.

# 3 Low-shot Categorization Using Mutual Exclusivity Bias (LSME)

## 3.1 Task Formalization

Consider an RGB scene containing multiple objects, including known instances and a single novel instance that lacks any explicit localization information and a word for its label. Our task can be partitioned into solving three well-known subtasks:

**Object Localization.** The first step is to localize the objects in an RGB scene.

**Open-world Recognition.** The second step is to differentiate between known and novel categories. The objective is to identify the instance within each scene that belongs to a novel category. To enable this, we assume the availability of a collection of labelled images from known categories, referred to as the *base* classes. Once the novel object identified it is associated with the novel word label, which we refer to as a *support* object.

**Low-shot Learning.** The last step is low-shot generalization. Given one or a few *query* images of objects from the novel categories, the goal is to correctly classify them based on the support objects.

**Data Generation Pipeline.** Given any 3D categorical dataset, following the convention of low-shot learning, we first partition these object categories into disjoint sets: base classes and low-shot classes. While base classes are commonly used to learn robust feature representations, low-shot classes are for evaluating the generalization ability of the models in a low data setting.

*Data Rendering.* To generate each scene, we first randomly select a subset of objects. The rotational poses for the objects are obtained using rigid body simulation. The objects are scaled and placed into the scene at random locations making sure collisions do not occur. Scenes are generated with varying background, illumination, and camera viewpoint. Please refer to the Supplement for more detailed descriptions of the generating process.

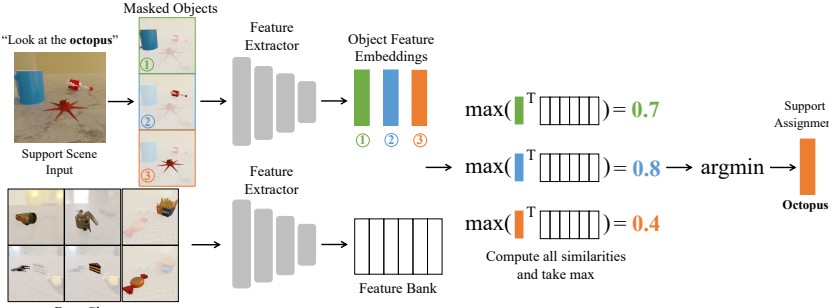

Figure 5: The process of Mutual Exclusivity Bias in our models. The input is the support scene with a novel word label. The model first needs to 1) localize the objects in the scene, 2) extract feature embeddings for these objects and compare with the features of the objects from the base classes using cosine distance, 3) take the maximum similarity score between each object embedding and the objects in the feature bank, 4) determine the unknown object by taking the object with the minimum similarity score and then assign the novel word label to this object.

## 3.2 Data Variants

To support a comprehensive analysis and gain insights into the limitations of different methods, we generate data with increasing difficulty levels of variability (please see Fig. 4).

The first group of variants considers scenes with a single object (-SObj) (left side of Fig. 4). The simplest setting concerns object instance recognition (Inst-SObj)—recognizing the same object based on a few images in the same pose. Our second group of variants requires models to generalize to different instances of the same category (Categ-SObj), while our third setting also takes into consideration random initial object pose sampling (Categ-SObj-PoseVar). Note that Categ-SObj is similar to the standard low-shot learning setting [44, 43]. These settings do not require mutual exclusivity bias.

The second group of has multiple objects (-MObj) (right side of Fig. 4). For the first, labels for the shots are given (Categ-MObj). This is similar to Categ-SObj-PoseVar, except the objects might be partially occluded. The second variant requires using mutual exclusivity bias to assign the label to the correct object before low-shot generalization (Categ-MObj-SuppAssign). Finally, LSME also requires object localization.

Note that while the first data variant (Inst-SObject) yields a straightforward task that does not require category generalization, all subsequent variants (Categ-SObject, Categ-SObject-PoseVar, and Categ-MObject) require instance-to-category generalization, as in the standard low-shot learning setting. Although not requiring mutual exclusivity bias, these variants are significantly more challenging in comparison to the standard low-shot learning formulation, because they introduce pose variability and include multiple objects, as described in Sec. 4.

## 3.3 Our Approach

**Representation Learning.** Motivated by the understanding that multi-view observations provide rich visual information about the 3D environment that aids downstream tasks [43, 17], we adopt a strategy of pretraining the feature extractor by leveraging contrastive learning on multi-view scenes. Unlike most self-supervised approaches that primarily operate on the scene level even when the scene consists of multiple objects, we focus on the object-level representation learning. Specifically, given scenes of multiple objects captured from various viewpoints, along with the corresponding instance masks for each object, we learn to match the representation of the same object across different views.

We use pre-trained backbones, (e.g. DINOv1 [5], DINOv2 [33]) contrastive training strategy with a momentum encoder [15]. Given two views of the same scene, $v_1$ and $v_2$, we first use the instance mask associated with each object in the scene to eliminate the background and other objects. Subsequently, we extract the query object feature by performing a forward pass of the image encoder on $v_1$. For each query feature, we minimize the InfoNCE [32] loss function. The positive sample is the feature of the same object in $v_2$ while the negative set consists of object features from the memory queue as in MoCo-v2 [8] and different objects from the same scene. This approach is inspired by the VISPE++

Table 2: Low-shot recognition on the Toys4k dataset in the single object setting. All methods consistently drop in accuracy when evaluated on the harder data variants.

| Variants | DINOv1-S/8 | | DINOv2-S/14 | | DINOv2-B/14 | |
|---|---|---|---|---|---|---|
| | 1-shot 5-way | 1-shot 10-way | 1-shot 5-way | 1-shot 10-way | 1-shot 5-way | 1-shot 10-way |
| Inst-SObj | 95.80 | 92.37 | 95.75 | 93.06 | 96.50 | 94.22 |
| Categ-SObj | 73.06 | 60.73 | 77.11 | 66.62 | 79.69 | 69.55 |
| Categ-SObj-PoseVar | 68.84 | 57.45 | 73.07 | 61.44 | 75.18 | 66.30 |

Table 3: Results on low-shot recognition on the Toys4k dataset in multi-object setting. All methods consistently experience a significant drop in low-shot accuracy when mutual exclusivity is required, and further decrease when instance segmentation is involved.

| Variants | DINOv1 ViT S/8 | | DINOv2 ViT S/14 | | DINOv2 ViT B/14 | | CLIP-Img ViT B/16 | | ImageBind ViT H/16 | |
|---|---|---|---|---|---|---|---|---|---|---|
| | LSA | SA | LSA | SA | LSA | SA | LSA | SA | LSA | SA |
| Categ-MObj | 56.99 | N/A | 56.95 | N/A | 57.92 | N/A | 56.76 | N/A | 60.49 | N/A |
| Categ-MObj -SuppAssign | 40.21 | 51.68 | 41.26 | 52.28 | 43.21 | 54.96 | 41.22 | 51.64 | 45.91 | 58.58 |
| LSME | 36.44 | 46.92 | 37.08 | 48.16 | 39.24 | 50.88 | 38.25 | 48.96 | 38.85 | 50.24 |

baseline in [43]. For each input view pair, we ensure to only train on objects that are visible in both views (e.g. with instance segmentation area greater than some threshold $\sigma$ pixels)

**Unsupervised Instance Segmentation.** We use a fine-tuned FreeSOLO [47] model as our segmenter, benefiting from its strong performance in instance segmentation tasks. Fine tuning is done on 1K scenes from the ABC dataset, which have been annotated with instance segmentation masks.

**Low-shot Learning.** We apply the following support assignment and low-shot generalization technique in all methods.

*Support Assignment:* We first collect a feature library consisting of object features extracted from the base classes as learned objects. During the low-shot training phase, for each input scene, our goal is to determine which object should be assigned the novel label. For every object in the scene, we calculate its similarity with each object in the feature library using cosine distance. To determine the final similarity score with the feature library, for each object in the scene, we select the maximum similarity measure across all objects in the library. Finally, the object in the scene with the lowest similarity score is assigned the novel label. Under this procedure, a learning agent would associate the novel label to the object that it is least familiar with (Fig. 5).

*Low-shot Generalization:* Given a query scene, we compute the cosine distance between each object in the query scene and every support object identified in the support assignment phase. We then determine the nearest neighbor for each query object within the support set and assign the label of the query object to its closest counterpart in the support set.

## 4 Experiments

### 4.1 Evaluation Setup

**Dataset.** Similar to prior works [44, 43] and the convention in low-shot learning, we partition the categories of ShapeNetCore.v2 [6] and Toys4k [44]. These datasets are divided into two disjoint subsets: base classes, for learning object-label associations, and test classes, for evaluating low-shot object recognition performance. We generate 1K scenes each for support and query sets, and 500 base scenes for multi-object experiments. Each scene in our dataset consists of 3 objects rendered in 20 views. To study mutual exclusivity bias, we ensure that each scene in the support set only contains 1 object from a novel category while the remaining objects come from base classes. We further incorporate a subset of CO3D [38] that shares 13 categories with the test classes in Toys4k to investigate the effectiveness of occlusion-aware feature representations in the real-world scenario. Since CO3D is an object-centric dataset, it is not possible to directly test LSME on CO3D. We only evaluate this dataset on the standard low-shot setting where mutual exclusivity bias is not considered.

We pre-train our models using the 3D object instance dataset ABC [21] without any category structure. We use a subset of ABC that consists of 100K objects. We render 10K scenes with 20 views per scene (8K for training and 2K for validation), each containing 2-3 objects with random object poses

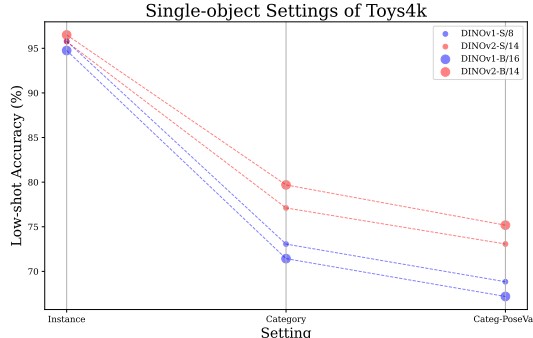
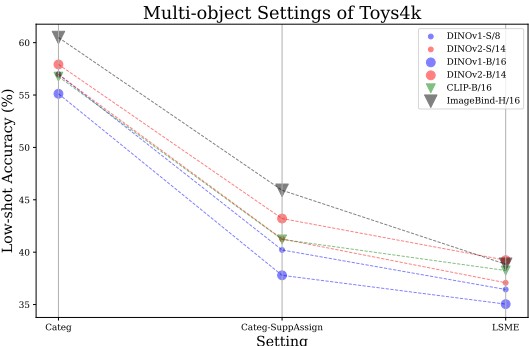

Figure 6: Low-shot accuracy of DINOv1 and DI-NOv2 pre-trained models on the Single-object settings of Toys4k. The size of the markers corresponds to the number of parameters of the models.

Figure 7: Performance of pre-trained models on the Multi-object settings of Toys4k. Circle and triangle markers represent self-supervised vision models and models that have linguistic input respectively.

Table 4: Performance of different methods on Toys4k under Categ-SObj-PoseVar and Categ-MObj settings. These settings solve a similar problem, with Categ-MObj having object occlusions present in both support and query objects. Performance of all methods drops significantly when faced with occlusion cases.

| Method | DINOv1 S/8 | DINOv2 S/14 | DINOv2 B/14 |
|---|---|---|---|
| Categ-SObj-PoseVar | 68.84 | 73.07 | 75.18 |
| Categ-MObj | 56.99 | 56.95 | 57.92 |

and scales. We further randomize the textures and surface materials of the objects in each scene (refer to Fig. 3).

**Evaluation Protocols.** We evaluate the performance of the baselines using the following metrics: 1) support assignment accuracy (SA) which quantifies the percentage of accurately identifying the novel instance within the scene, and 2) low-shot accuracy (LSA) for measuring low-shot performance, and 3) mIoU for instance segmentation. In this work, we follow the standard framing of low-shot inference, such that "1-shot 5-ways" means that each episode has 5 novel categories, each with only 1 object during the low-shot training phase. Unless stated otherwise, the experiments in this paper are evaluated on the 1-shot-5-way setup with 500 episodes. We report the confidence intervals and experiments on other setups in the Supplement. We make sure that novel objects are clearly visible during both support assignment and low-shot generalization phases (e.g. with instance segmentation area greater than threshold $\sigma$ pixels).

**Representation Learning Baselines.** Our main design constraint is that pre-training prior to the low-shot phase cannot incorporate any object labels, in order to avoid any concerns of label leakage. We focus on analyzing the performance of models initialized from the SOTA self-supervised vision models DINOv1[5] and DINOv2 [33]. This allows us to gain further insights into whether LSME can be tackled without prior language inputs. We further investigated the performance of large-scale vision-language foundation models, in order to characterize the difference between self-supervised and weakly-supervised pretraining approaches. We present findings using CLIP [37] and ImageBind [12] pre-trained models. Note that these models violate the constraint by having unrestricted (in terms of category composition) image captions in their pre-training.

## 4.2 LSME and Related Tasks on Synthetic Data

We first analyze the performance of pre-trained representations on settings with varying difficulty, building up to LSME.

**Single Object Setting.** We present the results for the single object setting on Toys4k in Table 2 and Figure 6 on 1-shot 5-way and 1-shot 10-way set ups. We observe a decrease in performance when tested on the harder variants for all models. While the difference between Categ-SObj-PoseVar and Categ-SObj is not significant, we observe a more prominent gap between Categ-SObj and Inst-SObj. This indicates the challenge faced by the models when generalizing from instance to category level.

**Multiple Objects Setting.** We report these results in Table 3 and Figure 7. The performance of all models exhibits a significant decrease ($\sim 15\%$) when mutual exclusivity bias is required (2nd row).

Table 5: Performance of DINOv2 and our method fine-tuned on Toys4k and ABC on Toys4k under LSME setting. All methods use ViT B/14 as the backbone and our method is initialized with pretrained DINOv2 weights. Training on ABC improves the performance significantly, surpassing the model that was trained on the base classes of Toys4k with the same number of scenes.

| Method | LSA | SA |
|---|---|---|
| DINOv2 | 39.24 | 50.88 |
| Ours-DINOv2-Toys | 43.62 | 53.44 |
| Ours-DINOv2-ABC | **47.70** | **61.32** |

Table 6: Performance of our baseline finetuned with different backbones on Toys4k under LSME settings with four object segmenters. Best performance is highlighted in bold while underline represents second best performance. The quality of the instance masks plays a significant role in the low-shot and shot assignment performance for all methods.

| | mIoU | | Ours-DINOv1 S/8-ABC | | Ours-DINOv2 S/14-ABC | | Ours-DINOv2 B/14-ABC | |
|---|---|---|---|---|---|---|---|---|
| | Support | Query | LSA | SA | LSA | SA | LSA | SA |
| FreeSOLO [47] | 0.52 | 0.54 | 32.31 | 43.24 | 33.99 | 44.84 | 35.50 | 48.92 |
| CutLer [49] | 0.61 | 0.63 | 34.62 | 42.40 | 36.34 | 46.08 | 39.42 | 52.04 |
| SAM [20] | 0.72 | 0.73 | 35.72 | 48.96 | 38.58 | 52.04 | 42.38 | 56.92 |
| FreeSOLO-ABC | **0.81** | **0.83** | **41.19** | **55.40** | **43.92** | **57.88** | **47.70** | **61.32** |

Table 7: Performance of DINOv2 and ours using ViT B/14 at LSME on ShapeNetCore.v2. Our method demonstrates an improvement in both low-shot and shot assignment accuracy.

| Method | LSA | SA |
|---|---|---|
| DINOv2 | 32.55 | 47.68 |
| Ours-DINOv2-ABC | 41.03 | 58.92 |

This decline can be attributed to the challenges associated with imperfect support assignment, which negatively affects the low-shot accuracy across all models.

*Effect of Segmentation Quality.* Table 6 provides additional evidence of the significance of the quality of the instance masks generated by the segmenter. It demonstrates that a decrease of $0.1$ in mIoU (between the last 2 rows) can result in a substantial impact on performance, with approximately a $6 - 7\%$ decrease in low-shot and support assignment accuracy. This emphasizes the critical role that accurate instance masks play in achieving high-performance results in both LSA and SA metrics.

*Effect of Occlusion.* Table 3 highlights the impact of occlusion caused by other objects in the scene on the low-shot performance of the models. While Categ-SObj-PoseVar and Categ-MObj address a similar problem, Categ-MObj considers potential occlusion in both support and query objects due to multiple objects being in the scene. We observe a decrease in performance for all models when occlusion is present. Specifically, DINOv1 experiences an average decrease of 11.85%, while DINOv2 models show greater decreases of 16.12% and 17.26% respectively.

These findings indicate the challenge posed by occlusion in low-shot learning scenarios. The presence of occlusion introduces additional complexity and ambiguity, making it more difficult for the models to accurately associate novel objects with their corresponding categories. This highlights the importance of developing methods purpose-built for the task of low-shot learning with mutual exclusivity.

**Contrastive Finetuning on ABC Improves Performance.** In Table 5 we show the advantage of contrastive finetuning on the ABC [21] dataset. While prior work [43] has demonstrated the benefits of pretraining feature representations on ABC, we are the first to show the advantages of ABC in a multi-object setting. Additionally, we train our model on an equal number of scenes from the base classes of Toys4k. Although this approach enhances the low-shot generalization performance on the test classes of Toys4k compared to DINOv2, its performance is inferior to the model trained on ABC. This highlights the benefits of leveraging ABC in the context of multi-object understanding and its potential for improving the capabilities of models in complex scenarios.

In addition, the performance comparison between DINOv2 with ViT B/14 backbone and our model fine-tuned on ABC is presented in Tab. 7 on the ShapeNetCore.v2 dataset. Notably, our model consistently outperforms DINOv2 across both the LSA and SA metrics, showcasing an improvement of approximately 10%.

## 4.3 Low-shot Generalization on CO3D Dataset

We investigate the generalization capabilities of pretraining on the multi-object multi-view ABC dataset, extending beyond the synthetic data domain to real-world datasets. Considering the negative impact of object occlusions and incomplete object masks on the performance of models, we hypothesize that pretraining on ABC scenes with multiple objects where occlusions are present improves the models' ability to handle such scenarios. To test this hypothesis, we conduct an experiment where we randomly mask the instance segmentations and evaluate the models on these masked segmentations in the standard low-shot setting. Figure 8 showcases the performance of different approaches

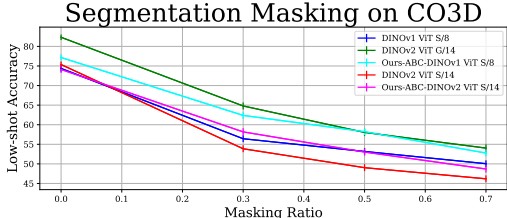

Figure 8: Performance of different methods on the segmentation masking experiment on CO3D. Low-shot performance of these methods decreases as the mask ratio increases. Models pretrained on ABC exhibit a slower decrease rate.

on CO3D dataset with varying mask ratios. The models finetuned on ABC exhibit a better performance compared to other models as the mask ratios increase. Even at a mask ratio of 0.5, our ABC-finetuned model with DINOv1 ViT S/8 pretrained weights performs on par with DINOv2 ViT G/14. Note that DINOv2 ViT G/14 has significantly more parameters and was trained on an order of magnitude more data than our model. These results demonstrate the benefit of occlusion-aware feature representations.

## 4.4 Limitations & Future Work

One important limitation of our work is the absence of uncertainty reasoning. In real-world scenarios, agents often encounter multiple unknown objects, requiring the integration of novel labels acquired in diverse contexts to correctly associate objects with their corresponding labels. This more complex setting requires continuous integration of new information and reasoning in ambiguous situations. Another challenging real-life infant-learning scenario is where objects might have multiple names (e.g., "dog" and "husky" both referring to the breed "husky"). Due to the procedural nature of our data generation system, we have the ability to increase the task complexity as solutions to LSME improve beyond the current baselines. Future research can explore more challenging settings and developing approaches that incorporate uncertainty reasoning to improve performance in ambiguous scenarios.

**Negative Societal Impact.** Training and evaluating large-scale self-supervised learning models as well as generating data require extensive GPU usage, which negatively impacts the environment. Advancements in hardware design and techniques for optimizing deep models offer potential solutions to mitigate this impact.

## 5 Conclusion

We present a novel setting called Low-shot Object Learning with Mutual Exclusivity Bias (LSME), which requires comprehensive reasoning about scenes with complex object interactions. We conduct a thorough analysis of the challenges present in LSME and their impact on SOTA models by generating various problem variants. Based on these insights, we propose a pretraining strategy that outperforms SOTA baselines on both synthetic and real-world data. Additionally, we release our open-source data generation pipeline and the generated datasets for further research.

## Broader Impact

The challenging LSME task provides a framework for devising algorithms that can learn from limited data input. We hope that LSME is an example computational framework that can serve as a template to formulate learning tasks based on other insights from the developmental psychology community. Future development of this setting can provide a learning environment that closely resembles real-world scenarios, effectively for further studies of both computer vision and developmental psychology communities.

## Acknowledgement

This work was supported by NIH R01HD104624-01A1.

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

# APPENDIX

This appendix is structured as follows: We first provide more details about data information in Section A. We then show additional results in Section B. Finally, we provide additional training details about our baseline models in Section C.

## A Data

### A.1 Datasets

In our work, we performed experiments and analysis using three datasets: Toys4K [44], ShapeNet-Core.v2 [6], ABC [21], and CO3D [38]. In the following section, we provide comprehensive details about each of these datasets.

**Toys4K [44].** This dataset consists of 4,179 object instances in 105 categories. We use the base and low-shot splits provided by Stojanov et al. [44]. In particular, the base classes consist of 40 categories while the low-shot classes have 55 categories. Objects in this dataset were collected under Creative Commons and royalty-free licenses. (Please refer to Table 8 for base/low-shot split compositions).

**ShapeNetCore.v2 [6].** This dataset consists of 52K objects in 55 categories. We partition these categories into 25 base and 30 low-shot classes (see Table. 8). The terms of use for ShapeNet are specified on their website, which can be accessed at https://shapenet.org/terms.

**ABC [21].** For pretraining our representation learning models, we used a subset of 100K object instances from ABC, which contains a total of 750K instances. Note that this dataset lacks categorical structures. The dataset is distributed under the MIT license. More licensing information is available at https://deep-geometry.github.io/abc-dataset/#license.

**CO3D [38].** We chose the 13 classes out of 51 classes that overlap with Toys4K for low-shot validation, detailed in Table 8. The terms of use for CO3D are specified at https://ai.facebook.com/datasets/co3d-downloads/.

### A.2 Data Generation

**Software.** We used Blender 2.93 [1] with ray-tracing renderer Cycles for data generation and rendering.

**Assets.** Objects are placed on top of a plane that simulates the ground/floor with PBR materials and image-based lighting from HDRI environment maps are used to illuminate scenes. We collected these assets from PolyHaven [2]. The list of assets used is shown in Table 9.

**Scene Generation.** Given any 3D categorical dataset, we first partition these object categories into disjoint sets: base classes and low-shot classes. For each object in the dataset, we preprocess it by simulating a rigid body drop using Blender [1]. This simulation process is repeated 16 times, allowing us to collect metadata and initial rotational poses for each object. These collected data are used in the subsequent stages of scene generation.

To generate each scene, we first choose a subset of objects from the dataset. Their initial rotational poses are determined by randomly choosing from the preprocessed poses. Objects are then scaled and placed into the scene at random locations. We ensure that collisions do not occur by maintaining a minimum margin of $\Delta > 0$ between each pair of objects. We randomize the scene background by randomly choosing a pair of PBR material and HDRI environment map from the assets.

**Data Rendering.** To render each view of the scenes, we first determine the camera position. The camera's position in the scene is specified by three parameters: $\theta \in [0, 2\pi]$, $r \in [r_{min}, r_{max}] > 0$, and $z \in [z_{min}, z_{max}] > 0$ where $\theta$ is the rotational angle, $r$ is the distance from the origin in the XY-plane, and $z$ denotes the world Z-coordinate of the camera. Note that $r_{min}, r_{max}, z_{min}, z_{max}$ are preset parameters. The world coordinate of the camera is computed by $(r\cos(\theta), r\sin(\theta), z)$. To determine the camera's orientation, it is set to point towards a location on the XY-plane that is within a small distance $\epsilon$ from the mean locations of the objects in the scene. This is done by rotating the camera in the world XY and YZ-planes. We then randomize illumination intensity, consistently for all the views of each scene.

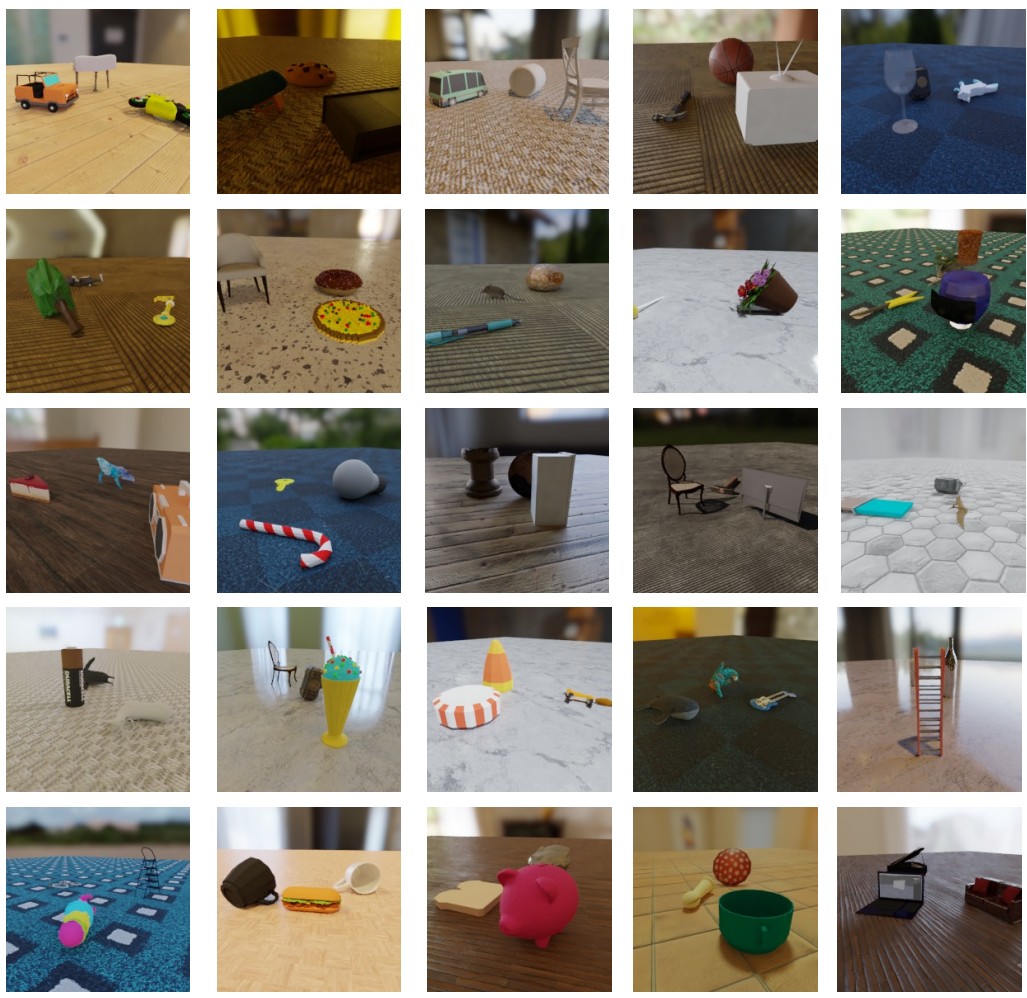

Figure 9: Rendered scenes for LSME on Toys4k [44]

**Generated Data for LSME.** We generated 1K scenes for each of support and query sets, with each scene consisting of 20 views. The data generated for LSME evaluation can be found at https://tinyurl.com/3a9r83z9. Additionally, the code for data generation is available on our GitHub repository at https://github.com/rehg-lab/LSME. Detailed parameters for scene generation can be found in Table 10.

### A.3 Data Augmentation for Contrastive Training

To augment the data, we applied various transformations, including random horizontal flips and brightness and color jittering. Following [43], we employed random object masking, where the object instance mask was used to eliminate the background. Additionally, we applied rotations and translations to the foreground object and incorporated background randomization techniques.

### A.4 More Data Visualizations

Figure 9 showcases additional examples of rendered scenes from the Toys4K dataset [44]. These examples highlight the diversity found in the background, illumination conditions, and object poses within the scenes.

In Figure 10, we demonstrate the instance mask prediction of the FreeSOLO [47] model finetuned on 1K scenes of ABC. The quality of the predicted masks is essential to solving LSME.

| ShapeNetCore.v2 | | Toys4k | | CO3D |
|---|---|---|---|---|
| Base | Low-shot | Base | Low-shot | Low-shot |
| chair | piano | candy | boat | TV |
| table | train | flower | lion | mouse |
| bathtub | file | dragon | whale | car |
| cabinet | pistol | apple | cupcake | toaster |
| lamp | motorcycle | guitar | train | microwave |
| car | printer | tree | pizza | donut |
| bus | mug | glass | marker | orange |
| cellular | rocket | cup | cookie | sandwich |
| guitar | skateboard | pig | sandwich | bicycle |
| bench | bed | cat | octopus | banana |
| bottle | ashcan | chair | monkey | bowl |
| laptop | washer | ice-cream | fries | motorcycle |
| jar | bowl | hat | violin | pizza |
| loudspeaker | bag | deer mouse | mushroom | |
| bookshelf | mailbox | penguin | closet | |
| faucet | pillow | ball | tractor | |
| vessel | earphone | fox | submarine | |
| clock | camera | dog | butterfly | |
| airplane | basket | knife | pear | |
| pot | remote | laptop | bicycle | |
| rifle | stove | pen | dolphin | |
| display | microwave | mug | bunny | |
| knife | microphone | plate | coin | |
| telephone | cap | chess piece | radio | |
| sofa | dishwasher | cake | grapes | |
| | keyboard | frog | banana | |
| | tower | ladder | cow | |
| | helmet | keyboard | donut | |
| | birdhouse | sofa | stove | |
| | can | trashcan | sink | |
| | | dinosaur | orange | |
| | | bottle | saw | |
| | | elephant | chicken | |
| | | pencil | hamburger | |
| | | key | piano | |
| | | monitor | light bulb | |
| | | hammer | spade | |
| | | screwdriver | crab | |
| | | robot | sheep | |
| | | bread | toaster | |
| | | | lizard | |
| | | | motorcycle | |
| | | | mouse | |
| | | | pc mouse | |
| | | | bus | |
| | | | helicopter | |
| | | | microwave | |
| | | | cell battery | |
| | | | drum | |
| | | | panda | |
| | | | TV | |
| | | | car | |
| | | | helmet | |
| | | | fridge | |
| | | | bowl | |

Table 8: Split composition of ShapeNetCovre.v2, Toys4K and CO3D

# B   Additional Experiments

## B.1   Evaluation Metric Details

We evaluate the performance of the baselines using the following metrics: 1) support assignment accuracy (SA) which quantifies the percentage of accurately identifying the novel instance within the scene, and 2) low-shot accuracy (LSA) for measuring low-shot performance, and 3) mean

| PBR | HDRI |
|---|---|
| Carpet001 | Aft Lounge |
| Carpet005 | Anniversary Lounge |
| Carpet006 | Balcony |
| Carpet007 | Cabin |
| Carpet008 | Cayley Interior |
| Carpet009 | Children's Hospital |
| Carpet013 | Colorful Studio |
| Carpet014 | Entrance Hall |
| Fabric024 | Fireplace |
| Fabric025 | Hotel Room |
| Fabric028 | Kiara Interior |
| Marble012 | Lapa |
| Planks001 | Lebombo |
| Planks009 | Lythwood Lounge |
| Planks011 | Lythwood Room |
| Planks013 | Moonlit Golf |
| Planks014 | Music Hall |
| Planks018 | Photo Studio |
| Terrazzo001 | Reading Room |
| Tiles001 | Roof Garden |
| Tiles027 | Small Empty House |
| Tiles071 | Spiaggia Di Mondello |
| Tiles072 | St Fagans Interior |
| WoodFloor005 | Umhlanga Sunrise |
| WoodFloor028 | Wooden Lounge |

Table 9: List of assets used in data generation.

| Parameter | Value |
|---|---|
| Camera $r$ | $[1.0, 1.1)$ |
| Camera $z$ | $[0.3, 0.5)$ |
| Camera jittering $\epsilon$ | 0.01 |
| Object scale | $[0.35, 0.45)$ |
| Object location | $[-0.5, 0.5)$ |
| Illumination intensity | $[0.6, 0.8)$ |
| Object margin $\Delta$ | 0.4 |

Table 10: Data rendering parameters.

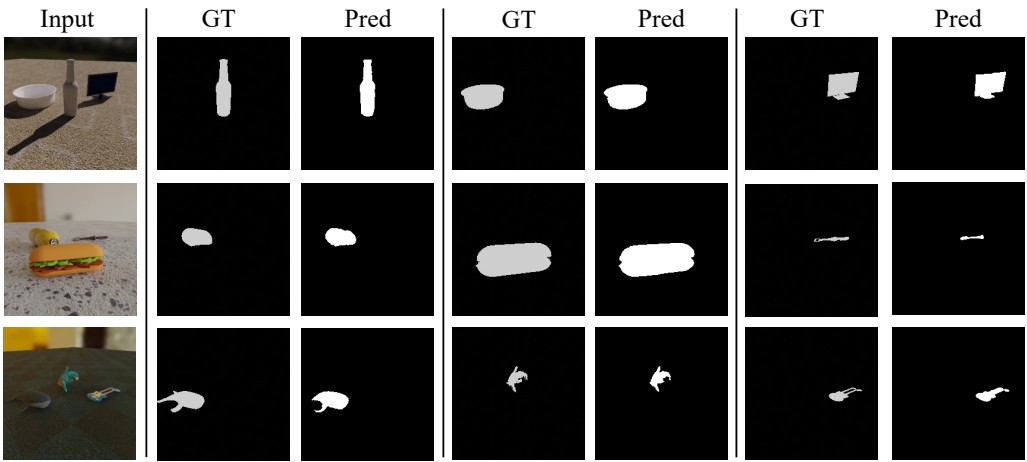

Figure 10: Segmentation prediction results on Toys4K [44] using FreeSOLO [47] fine-tuned on ABC model

intersection-over-union (mIoU) for instance segmentation as detailed below. For each episode,

$$SA = \frac{1}{N_s} \sum_{i=1}^{N_s} \mathbb{1}\{\hat{o}_i = o_i\}$$

Table 11: Results on low-shot recognition on the Toys4k dataset in single object setting. All methods consistently experience a significant drop in accuracy when being evaluated on the harder data variants.

| Variants | DINOv1-S/8 | | DINOv2-S/14 | | DINOv2-B/14 | |
|---|---|---|---|---|---|---|
| | 1-shot 5-way | 1-shot 10-way | 1-shot 5-way | 1-shot 10-way | 1-shot 5-way | 1-shot 10-way |
| Inst-SObj | $95.80_{\pm0.46}$ | $92.37_{\pm0.42}$ | $95.75_{\pm0.44}$ | $93.06_{\pm0.41}$ | $96.50_{\pm0.43}$ | $94.22_{\pm0.37}$ |
| Categ-SObj | $73.06_{\pm0.96}$ | $60.73_{\pm0.76}$ | $77.11_{\pm0.89}$ | $66.62_{\pm0.78}$ | $79.69_{\pm0.99}$ | $69.55_{\pm0.77}$ |
| Categ-SObj-PoseVar | $68.84_{\pm1.04}$ | $57.45_{\pm0.77}$ | $73.07_{\pm1.03}$ | $61.44_{\pm0.80}$ | $75.18_{\pm1.04}$ | $66.30_{\pm0.79}$ |

Table 12: Results on low-shot recognition on the Toys4k dataset in multi-object setting. All methods consistently experience a significant drop in low-shot accuracy when mutual exclusivity is required, and further decrease when instance segmentation is involved.

| Variants | DINOv1 ViT S/8 | | DINOv2 ViT S/14 | | DINOv2 ViT B/14 | | CLIP-Img ViT B/16 | | ImageBind ViT H/16 | |
|---|---|---|---|---|---|---|---|---|---|---|
| | LSA | SA | LSA | SA | LSA | SA | LSA | SA | LSA | SA |
| Categ-MObj | $56.99$ $\pm0.97$ | N/A | $56.95$ $\pm0.99$ | N/A | $57.92$ $\pm1.04$ | N/A | $56.76$ $\pm1.01$ | N/A | $60.49$ $\pm1.00$ | N/A |
| Categ-MObj -SuppAssign | $40.21$ $\pm1.10$ | $51.68$ $\pm1.95$ | $41.26$ $\pm1.15$ | $52.28$ $\pm1.86$ | $43.21$ $\pm1.21$ | $54.96$ $\pm1.89$ | $41.22$ $\pm1.16$ | $51.64$ $\pm1.87$ | $45.91$ $\pm1.25$ | $58.58$ $\pm2.00$ |
| LSME | $36.44$ $\pm1.08$ | $46.92$ $\pm2.04$ | $37.08$ $\pm1.05$ | $48.16$ $\pm1.87$ | $39.24$ $\pm1.17$ | $50.88$ $\pm1.91$ | $38.25$ $\pm1.14$ | $48.96$ $\pm2.03$ | $38.85$ $\pm1.14$ | $50.24$ $\pm1.98$ |

Table 13: Performance of DINOv2 and our method fine-tuned on Toys4k and ABC on Toys4k under LSME setting. All methods use ViT B/14 as the backbone and our method is initialized with pretrained DINOv2 weights. Training on ABC improves the performance significantly, surpassing the model that was trained on the base classes of Toys4k with the same number of scenes.

| Method | LSA | SA |
|---|---|---|
| DINOv2 | $39.24_{\pm1.17}$ | $50.88_{\pm1.91}$ |
| Ours-DINOv2-Toys | $43.62_{\pm1.29}$ | $53.44_{\pm1.89}$ |
| Ours-DINOv2-ABC | $\mathbf{47.70_{\pm1.26}}$ | $\mathbf{61.32_{\pm1.86}}$ |

where $o$, $\hat{o}$, and $N_s$ are ground truth object, predicted object, and the number of support objects respectively (e.g. in the 1-shot-5-way setup $N_s = 5$ since there are 5 support objects in the episode.)

$$LSA = \frac{1}{N_q} \sum_{i=1}^{N_q} \sum_{k=1}^{N_w} \mathbb{1}\{\hat{y}_{ik} = y_{ik}\}$$

where $\hat{y}$ and $y$ are predicted and ground truth labels respectively. The number of query objects is denoted as $N_q$ while $N_w$ is the number of classes (e.g. in the 1-shot-5-way setup, $N_w = 5$ since there are 5 novel classes.)

$$mIoU = \sum_{i=1}^{N} \frac{\hat{m}_i \cap m_i}{\hat{m}_i \cup m_i}$$

where $m$, $\hat{m}$, and $N$ denote the ground truth mask, predicted mask, and number of objects respectively.

## B.2 Main Manuscript Results

In this section, we report the confidence intervals of the experiment results in the main manuscript (Please see Tables 11, 12, 13, 14, and 15). We evaluate our models with 500 episodes and 15 query scenes for each episode.

## B.3 Other Low-shot Setups

Table 16 presents the results of DINOv2 ViT B/14, both pre-trained and fine-tuned on ABC, in various low-shot setups, including 1-shot-5-way, 5-shot-5-way, 1-shot-10-way, and 5-shot-10-way under LSME setting on Toys4k.

While the support assignment accuracy (SA) remains consistent across all low-shot setups, the low-shot accuracy shows a notable improvement in the 5-shot scenarios with an approximate 16% increase in low-shot accuracy in both 5-way and 10-way setups.

Table 14: Performance of different methods on Toys4k under Categ-SObj-PoseVar and Categ-MObj settings. These settings solve a similar problem, with Categ-MObj having object occlusions present in both support and query objects. Performance of all methods drops significantly when faced with occlusion cases.

| Method | DINOv1 S/8 | DINOv2 S/14 | DINOv2 B/14 |
|---|---|---|---|
| Categ-SObj-PoseVar | 68.84 ±1.04 | 73.07 ±1.03 | 75.18 ±1.04 |
| Categ-MObj | 56.99 ±0.97 | 56.95 ±0.99 | 57.92 ±1.04 |

Table 15: Performance of our baseline finetuned with different backbones on Toys4k under LSME settings with four object segmenters. Best performance is highlighted in bold while underline represents second best performance. The quality of the instance masks plays a significant role in the low-shot and shot assignment performance for all methods.

| | mIoU | | Ours-DINOv1 S/8-ABC | | Ours-DINOv2 S/14-ABC | | Ours-DINOv2 B/14-ABC | |
|---|---|---|---|---|---|---|---|---|
| | Support | Query | LSA | SA | LSA | SA | LSA | SA |
| FreeSOLO [47] | 0.52 | 0.54 | 32.31 ±0.93 | 43.24 ±1.94 | 33.99 ±0.95 | 44.84 ±1.95 | 35.50 ±0.99 | 48.92 ±1.88 |
| CutLer [49] | 0.61 | 0.63 | 34.62 ±0.87 | 42.40 ±1.96 | 36.34 ±0.96 | 46.08 ±1.96 | 39.42 ±1.04 | 52.04 ±1.97 |
| SAM [20] | 0.72 | 0.73 | 35.72 ±1.06 | 48.96 ±2.06 | 38.58 ±1.10 | 52.04 ±2.01 | 42.38 ±1.17 | 56.92 ±1.90 |
| FreeSOLO-ABC | **0.81** | **0.83** | **41.19** ±1.16 | **55.40** ±2.03 | **43.92** ±1.23 | **57.88** ±1.97 | **47.70** ±1.26 | **61.32** ±1.86 |

Table 16: Results on low-shot recognition on the Toys4k dataset in multi-object setting. All methods consistently experience a significant drop in low-shot accuracy when mutual exclusivity is required, and further decrease when instance segmentation is involved.

| | DINOv2 ViT B/14 | | DINOv2 ViT B/14-ABC | |
|---|---|---|---|---|
| Low-shot Setup | LSA | SA | LSA | SA |
| 1-shot-5-way | 39.24±1.17 | 50.88±1.91 | 47.70±1.26 | 61.32±1.86 |
| 5-shot-5-way | 55.03±0.99 | 50.22±0.99 | 63.52±1.02 | 60.60±1.13 |
| 1-shot-10-way | 28.32±0.73 | 51.32±1.46 | 35.66±0.82 | 61.10±1.30 |
| 5-shot-10-way | 43.26±0.70 | 50.62±0.69 | 51.72±0.75 | 60.85±0.74 |

## C   Models

**Representation Learning Models:** We use pre-trained backbones, (e.g. DINOv1 [5], DINOv2 [33]) contrastive training strategy with a momentum encoder[15]. Given two views of the same scene, $v_1$ and $v_2$, we first use the instance mask associated with each object in the scene to eliminate the background and other objects. Subsequently, we extract the query object feature by performing a forward pass of the image encoder on $v_1$. For each query feature, we minimize the InfoNCE [32] loss function.

$$\mathcal{L}_q = -\log \frac{\exp(q \cdot k_+/\tau)}{\exp(q \cdot k_+/\tau) + \sum_{k_-} \exp(q \cdot k_-/\tau)}$$

The positive sample $k_+$ is the feature of the same object in $v_2$ while the negative set $\{k_-\}$ consists of object features from the memory queue as in MoCo-v2 [8] and different objects from the same scene. For each input view pair, we ensure to only train on objects that are visible in both views (e.g. with instance segmentation area greater than some threshold $\sigma = 30$ pixels).

In our approach, we omit the projector and predictor components present in most contrastive learning approaches [15, 15, 7] since we found empirically that this gave better performance. We trained our model using AdamW optimizer with initial learning rate $5e^{-6}$ and weight decay 0, batch size 32 on 3 RTX 2080 GPUs for 50 epochs. Training took approximately 5 hours in clock time. Our pretrained weights can be found at https://tinyurl.com/3a9r83z9 and the training code is on our GitHub repository at https://github.com/rehg-lab/LSME. All pre-trained weights for other models are directly loaded from the corresponding released codebases.

**Segmentation Models:** We finetuned the pretrained FreeSOLO [47] model on 1K scenes of ABC dataset with instance mask annotations. To obtain the predicted instance masks for low-shot, we performed a forward pass of the fine-tuned model on our low-shot data. From the output masks,

we retained the ones with a confidence score above 0.5. To handle overlapping masks, we merged those with an IoU greater than 0.7. Finally, we employed the Hungarian matching algorithm [22] to associate each predicted mask with its corresponding ground truth mask. We finetuned FreeSOLO with batch size 6 on 3 RTX 2080 GPUs for 30K epochs.

