# OpenReview forum: "Low-shot Object Learning with Mutual Exclusivity Bias"
_NeurIPS.cc/2023/Track/Datasets_and_Benchmarks — NeurIPS 2023 Datasets and Benchmarks Poster_

### Official Review · Reviewer_HVNj · 2023-07-14
**A novel low-shot learning setting, but needs more thorough evaluation**

**Rating:** 6
**Confidence:** 4

**Strengths:**

- This work studies a novel low-shot learning setting, in which the learner needs to understand a new object concept with mutual exclusivity bias. A few scene images are given to the learner, which include both seen objects and unseen objects. The learner needs to reason the existence of previously not learned objects and build a connection between the unseen object and the new concept. This learning task presents a new challenge, as well as new opportunities for various real-world applications such as robot learning and continual learning.

- Various pre-trained visual representations are evaluated, including vision-only representations (e.g., DINO) and vision-language representations (e.g., CLIP). These different representations can help understand the role of language-based learning in gaining new object concepts.

- The proposed LSME solution is general and intuitive. It mainly includes three sub-solutions, object localization, open-world recognition, and low-shot learning. This framework can be extended when better solutions in these related topics are available.

- The writing is clear and easy to follow.

**Additional Feedback:**

None.

**Clarity:**

Yes, the paper is well written. One minor suggestion is to move the description of “Support Alignment”, “Low-shot Generalization”, and Figure 4 (Page 7) to Section 3.3 Our Approach.

**Correctness:**

The main conclusions in this work are sound. Please check “Opportunities For Improvement” for further suggestions on evaluation.

**Documentation:**

Yes.

**Ethics:**

No ethical concerns.

**Limitations:**

As mentioned in the submission, this work is somehow limited to one single unknown object in the scene image. Further technical improvement is required for extending LSME to multiple unknown objects.

**Opportunities For Improvement:**

- The evaluated methods are based on frozen, or further pre-trained visual representations. During the learning process where new objects and new concepts are given, the visual representations are not updated. In fact, low-shot generalization (to query images) is achieved via the nearest neighbor in the feature bank. This learning approach is not scalable, nor adaptive enough. Some potentially better baselines in few-shot object detection that can update the model may be considered, such as meta-learning [R1] and transfer learning [R2].

- The single-object settings (Inst-SObj, Categ-SObj, and Categ-SObj-PoseVar) seem to be not closely related to the main topic of this work. Solving such sub-tasks does not require using the mutual exclusivity bias.

- In addition to FreeSOLO, a more recent method CutLER [R3] has achieved stronger performance for unsupervised object detection and instance segmentation. The authors may consider including this method for better object localization.

- The pre-trained visual representations listed in the comparison (e.g., in Table 3) are not of the same scale, which makes it hard to understand the benefit of each pre-training strategy. It is suggested to use models with similar scales. For example, DINOv1 has pre-trained ViT-B/16, which fits better into Table 3.

[R1] Xiaopeng Yan, Ziliang Chen, Anni Xu, Xiaoxi Wang, Xiaodan Liang, Liang Lin. Meta R-CNN : Towards General Solver for Instance-level Few-shot Learning. In ICCV, 2019.

[R2] Xin Wang, Thomas E. Huang, Trevor Darrell, Joseph E. Gonzalez, Fisher Yu. Frustratingly Simple Few-Shot Object Detection. In ICML, 2020.

[R3] Xudong Wang, Rohit Girdhar, Stella X. Yu, Ishan Misra. Cut and Learn for Unsupervised Object Detection and Instance Segmentation. In CVPR, 2023.

**Relation To Prior Work:**

Yes, this work has details about the similarities and differences between the proposed new task and prior work.

**Summary And Contributions:**

This work introduces a novel task, Low-shot Object Learning with Mutual Exclusivity Bias (LSME). Inspired by humans’ early-stage visual learning, this new task aims to learn new concepts with mutual exclusivity bias. In particular, when a scene image containing multiple seen and unseen objects and a new concept are given to the learner, the mutual exclusivity bias builds a connection between the unseen object and the new concept. The low-shot learning setting adds to the task difficulty as well as practical impact. In addition to a general framework for LSME solution, this work also creates a new dataset via rendering 3D shapes, and a benchmark for evaluating various pre-trained visual representations.

---

> ### Author Response · Authors · 2023-08-21
> **To Reviewer HVNj - Answers to Opportunities For Improvement**
>
> ### Q1. "The evaluated methods are based on frozen, or further pre-trained visual representations. During the learning process where new objects and new concepts are given, the visual representations are not updated. In fact, low-shot generalization (to query images) is achieved via the nearest neighbor in the feature bank. This learning approach is not scalable, nor adaptive enough. Some potentially better baselines in few-shot object detection that can update the model may be considered, such as meta-learning [1] and transfer learning [2]."
>
> We present the performance of Fsdet [2] in the table below. As shown in their paper, Fsdet demonstrated a stronger performance than [1]. It's important to highlight that Fsdet was originally designed for standard low-shot object detection scenarios where mutual exclusivity bias isn't a requirement. For a relevant comparison since Fsdet cannot be applied directly on LSME, we evaluate the performance of Fsdet and our baselines in the most challenging scenario that does not require mutual exclusivity bias: Categ-MObj with object localization prediction. During low-shot inference, our evaluation focuses solely on novel classes. We’d like to emphasize that our proposed task is novel and fundamentally different from the conventional low-shot object detection task.
>
> We further note that the baseline we present is not the only approach to tackle LSME, but rather a relevant baseline built on the most powerful learning systems that we have to date. Future work can directly explore more scalable and adaptive methods in the context of LSME.
>
> | | Low-shot Accuracy |
> |--| :----: |
> | Fsdet [2] | 42.53 $\pm$ 1.74 |
> | Ours - DINOv1 ViT S/8 - ABC| 56.76 $\pm$ 1.02 |
> | Ours - DINOv2 ViT S/14 - ABC| 58.62 $\pm$ 1.05|
> | Ours - DINOv2 ViT B/14 - ABC | 61.68 $\pm$ 1.10|
>
> ### Q2. "The single-object settings (Inst-SObj, Categ-SObj, and Categ-SObj-PoseVar) seem to be not closely related to the main topic of this work. Solving such sub-tasks does not require using the mutual exclusivity bias."
>
> The goal of these experiments is to highlight multiple factors that make LSME a challenging task and the capabilities that models need to have to succeed at LSME. Further, these settings can be used to diagnose the limitations of models during development. For example, our results show that current SOTA methods experience difficulty when occlusions are introduced, while they appear to be less sensitive to various pose changes. We noticed that the heading of Section 4.2 can be confusing since some of the data variants do not require mutual exclusivity bias. We have changed this heading in the paper.
>
> ### Q3. "In addition to FreeSOLO, a more recent method CutLER [3] has achieved stronger performance for unsupervised object detection and instance segmentation. The authors may consider including this method for better object localization."
> We present the performance of CutLER and SAM in Table 8, alongside FreeSOLO and our FreeSOLO-ABC baseline. Performing object localization using SAM improves the performance on LSME for all backbones compared to CutLER and FreeSOLO.
>
>
> ### Q4. "The pre-trained visual representations listed in the comparison (e.g., in Table 3) are not of the same scale, which makes it hard to understand the benefit of each pre-training strategy. It is suggested to use models with similar scales. For example, DINOv1 has pre-trained ViT-B/16, which fits better into Table 3."
>
> In Figures 6 and 7 we show the low-shot accuracy of different models with respect to their scales on single-object and multi-object settings respectively. We have included DINOv1 ViT B/16 model in these figures as suggested by Reviewer HVNj for comparison. The performance of all models decreases as the settings get more challenging.

---

> > ### Comment · Reviewer_HVNj · 2023-08-30
> > **Follow-up**
> >
> > The authors’ response is greatly appreciated, which successfully addresses most of my initial concerns. The additional results on DINO backbones and newer object localization models have convinced me to elevate my rating for this work.
> >
> > Meanwhile, I agree with Reviewers FRCv and xYTB that the problem settings in this study might be overly simplified, especially when compared to real-world learning scenarios that involve mutual exclusivity bias. I hope that future research will explore more complex yet realistic tasks, particularly those involving multiple unseen objects or classes in one scene.

---

### Official Review · Reviewer_FRCv · 2023-07-21
**An interesting formulation of mutual exclusivity in few-shot vision-language learning**

**Rating:** 8
**Confidence:** 5
**Correctness:** Yes.
**Clarity:** Yes, the paper is well written.

**Strengths:**

- The paper effectively utilizes the concept of inductive biases, specifically the mutual exclusivity principle, that humans use to learn and recognize objects in their environment. This concept is cleverly framed into a computational problem (LSME) in the context of machine learning.

- The problem setting and the task are well-motivated and relevant. Modeling such inductive biases could be essential for improving machine learning models' abilities to understand and interact with their environment. The authors provide a clear explanation of the mutual exclusivity bias and how it can be applied to machine learning tasks, bridging the gap between cognitive science and artificial intelligence.

- The authors experiment with several state-of-the-art models and are candid about where these models fall short. They propose an alternate baseline that considers the shortcomings of existing models and provides improved performance.

**Additional Feedback:**

Typos and formatting:

1. line 167: add space before "[12]"
2. line 123: capitalization
3. line 58: add space before "Our"
4. table 4,5,67: "|" too long after Method
5. table 6: "38.932" -> "38.93" all results should have the same number of significant figures

**Documentation:**

Yes

**Ethics:**

No, there are no foreseeable ethical concerns.

**Limitations:**

Yes, the authors adequately addressed the limitations and potential negative societal impact of their work. No foreseeable negative societal impact.

**Opportunities For Improvement:**

- The study assumes that each unfamiliar object will have a unique label, which might not always be the case in a real-world learning scenario, where multiple new objects may share a common label or vice versa. Meanwhile, the labels are familiar words (instead of tufa/dax words), so if the model has enough knowledge from pretraining, it may not demonstrate the capability of real few-shot learning.

- The LSME task setup, although inventive, might overly simplify the process of learning object names, as it only associates labels with noticeable objects based on the novel category instance visibility. The evaluation is largely dependent on the quality of localization, which means performance could be compromised by poor object identification or scene understanding.

- The abstract reasoning capabilities of a toddler are simplified into recognition tasks, which may not encapsulate all aspects of a toddler's learning process. For instance, toddlers often leverage cross-situational, syntactic, and social cues, which are not considered in this work.

**Relation To Prior Work:**

Yes. And I suggest some related works can be added to the comparison:

[1] Agrawal, H., Meirom, E. A., Atzmon, Y., Mannor, S., & Chechik, G. (2021, December). Known unknowns: Learning novel concepts using reasoning-by-elimination. In Uncertainty in Artificial Intelligence (pp. 504-514). PMLR.

[2] Jiang, G., Xu, M., Xin, S., Liang, W., Peng, Y., Zhang, C., & Zhu, Y. (2023). MEWL: Few-shot multimodal word learning with referential uncertainty. In ICML.

[3] Hill, F., Tieleman, O., von Glehn, T., Wong, N., Merzic, H., & Clark, S. (2020, September). Grounded Language Learning Fast and Slow. In ICLR.

**Summary And Contributions:**

This paper introduces an interesting formulation of mutual exclusivity in few-shot vision-language learning by proposing the LSME benchmark. In LSME, there a three sub-tasks: object localization, open-vocabulary recognition, and few-shot classification. Experiments on multiple baseline methods show the difficulty of LSME. This work also proposed a valid pretraining method that can demonstrate good performance on several LSME tasks.

---

> ### Author Response · Authors · 2023-08-21
> **To Reviewer FRCv - Answers to Opportunities For Improvement**
>
> ### Q1.1 "The study assumes that each unfamiliar object will have a unique label, which might not always be the case in a real-world learning scenario, where multiple new objects may share a common label or vice versa."
> In our experiments, we assume that each object belongs to a single, unique category, as this forms the core principle of mutual exclusivity bias from the developmental literature [1,2]. We acknowledge that this is a simplification of real-world scenarios in which objects might have multiple names (e.g., "dog" and "husky" both referring to the breed "husky"). This issue is now discussed in our limitations section.
>
> [1] Markman, E. M., & Wachtel, G. F. (1988). Children's use of mutual exclusivity to constrain the meanings of words. Cognitive psychology, 20(2), 121-157.
>
> [2] Merriman, W. E., Bowman, L. L., & MacWhinney, B. (1989). The mutual exclusivity bias in children's word learning. Monographs of the society for research in child development, i-129.
>
> ### Q1.2. "Meanwhile, the labels are familiar words (instead of tufa/dax words), so if the model has enough knowledge from pretraining, it may not demonstrate the capability of real few-shot learning."
> With respect to the issue of pretraining and familiar words, we have provided further clarification in the paper. The main focus of LSME is learning about novel object-category association. To prevent leakage by which pre-training data might include labeled data from the low-shot categories (i.e. familiar words), we impose the following constraint: Pre-training prior to the low-shot phase cannot incorporate any object-label association data. This constraint further aligns with the investigation of a key question in developmental psychology: how can a child rapidly learn words in the absence of any prior experience with object semantics. We ensure that our models are pre-trained without access to any word labels. Nonetheless, it's worth noting that even when weakly-supervised with language input (as seen with CLIP and ImageBind), these models still struggle to perform well on our task.
>
> ### Q2. "The LSME task setup, although inventive, might overly simplify the process of learning object names, as it only associates labels with noticeable objects based on the novel category instance visibility. The evaluation is largely dependent on the quality of localization, which means performance could be compromised by poor object identification or scene understanding."
> Object localization serves as a sub-task of LSME that impacts the final performance due to the inter-connections among all sub-tasks. We believe that localization is as crucial to learning objects as categorization. Therefore, during data generation, we ensure that while the object localization sub-task is challenging, it does not dominate as the sole driving force behind the performance in LSME. Specifically, we make sure that novel objects are clearly visible during both the support assignment and low-shot inference phases, making it feasible for reasonable performance (mIoU > 0.7) by applying off-the-shelf class-agnostic localization algorithm (e.g. SAM). To understand the impact of object localization performance on the final low-shot performance, we evaluated the models in a setting where ground truth segmentations are provided during training and inference (results in Table 3 second row, also shown below). We find that this does not lead to significant improvements in performance.
>
> | | DINOv1 ViT S/8  | DINOv2 ViT S/14  | DINOv2 ViT B/14  | CLIP ViT B/16 | ImageBind ViT H/16 |
> |---| :---: | :----: | :---: | :-----: | :-----: |
> Categ-MObj-SuppAssign| 40.21 | 41.26 | 43.21 | 41.22 | 45.91 |
> LSME | 36.44 | 37.08 | 39.24 | 38.25 | 38.85 |
>
> ### Q3. "The abstract reasoning capabilities of a toddler are simplified into recognition tasks, which may not encapsulate all aspects of a toddler's learning process. For instance, toddlers often leverage cross-situational, syntactic, and social cues, which are not considered in this work."
> We agree that LSME does not encapsulate all aspects of infant learning. However, our goal is not to provide a comprehensive model of toddler learning (as discussed in our limitations section). Rather, our aim is to develop a new task that incorporates more realistic visual inputs than prior work [1] and generalizes classical few-shot learning paradigms to a more challenging setting inspired by child development. While LSME simplifies the learning process into localization and recognition, our findings demonstrate that even SOTA models struggle with this task. Solving LSME can therefore serve as a foundational step towards addressing more challenging learning scenarios that cover additional, more complex, aspects of infant learning (please see our response to Reviewer xYTB).
>
> [1] Vong, W. K., & Lake, B. M. (2022). Cross‐Situational Word Learning With Multimodal Neural Networks. Cognitive science, 46(4), e13122.

---

> > ### Author Response · Authors · 2023-08-21
> > **To Reviewer FRCv - Relation To Prior Work**
> >
> > We thank the reviewer for the feedback. We have incorporated these works in the Related Work section (Section 2).

---

> > > ### Comment · Reviewer_FRCv · 2023-08-28
> > >
> > > Thanks for the authors' response. My concerns are adequately solved. I recommend acceptance of this paper and would like to increase my rating by one.

---

### Official Review · Reviewer_xYTB · 2023-07-22
**An interesting idea but with limited contribution**

**Rating:** 7
**Confidence:** 3
**Correctness:** The claims made in the submission is …
**Clarity:** Yes, the paper is well writen.

**Strengths:**

The first to explore mutual exclusivity via Low-shot Object Learning with Mutual Exclusivity Bias(LSME);
Multiple baselines are evaluated;

**Additional Feedback:**

No Additional Feedback

**Documentation:**

Yes.

**Ethics:**

No ethical issues

**Limitations:**

Currently, there is only one novel object in a scene. This setting might be too easy and can not reflect the real world.

**Opportunities For Improvement:**

Currently, there is only one novel object in a scene. This setting may be too easy and useless.

**Relation To Prior Work:**

No Prior Work, this work proposed a new task.

**Summary And Contributions:**

1) The first to provide a computational framing of mutual exclusivity via Low-shot Object Learning with Mutual Exclusivity Bias(LSME);
2) A dataset generation pipeline that enables the creation of progressively more challenging LMSE tasks using any set of 3D models;
 3) Performance benchmarking of multiple baselines including foundation models on this novel task;
4) A novel baseline method that defines the SOTA on LMSE

---

> ### Author Response · Authors · 2023-08-21
> **To Reviewer xYTB - Answers to Opportunities For Improvement**
>
> ### Q1. “Currently, there is only one novel object in a scene. This setting may be too easy and useless.”
> We would like to emphasize that while our current task setup is a simplification in comparison to real-life infant-learning scenarios, it nonetheless provides an essential first step in developing computational models for LSME. Our current setting with one novel object in the scene is the simplest framing of LSME that captures the fundamental properties of the task, and our baseline model performance suggests that this task is already challenging for SOTA foundation models.
>
> An example of a more advanced setting is to introduce multiple unfamiliar objects in a scene in the low-shot training phase, requiring models to encode the inherent ambiguity present during learning. In this scenario, models capable of addressing the one-object setting can identify unambiguous moments during learning (i.e. when only one novel object exists in the scene) and use elimination across different scenes to tackle the object-label association problem. We consider such a generalization as an area of future work, particularly once the current LSME benchmark has become saturated. Further, because of our modular codebase, scaling LSME to more challenging settings as described is straightforward (e.g. one can easily introduce more objects in the scene during rendering).

---

> > ### Comment · Reviewer_xYTB · 2023-08-29
> >
> > Thanks for your reply. I will keep the score at 7.

---

### Official Review · Reviewer_BMRQ · 2023-07-23
**Meaningful task but should be refined.**

**Rating:** 6
**Confidence:** 5

**Strengths:**

1. The proposed task is meaningful for measuring generalization.
1. L98 The related settings paragraph provides clear distinction from the existing tasks.

**Additional Feedback:**

Typo
* L58 Missing space after a period.
* L154 The second group of [variants] has multiple objects.

**Clarity:**

The paper would be easier to read if it provides the definitions in the very first paragraph. E.g., `object learning`, `mutual exclusivity bias`, and `low-shot`.

**Correctness:**

L90 CLIP and ImageBind are not self-supervised but they are in the self-supervised learning paragraph. They require image-text pairs.

**Documentation:**

Yes. L136

**Ethics:**

No.

**Limitations:**

1. It mentions that the proposed task is far from the real-world scenario where multiple objects in a scene can be unfamiliar.

**Opportunities For Improvement:**

1. The proposed task has multiple sub-tasks that are not easily solvable for given the proposed dataset.
1. There are too many variables for the proposed task: segmentation and pretraining. The steps for the benchmark should be clearer. It is pretraining on ABC -> low-shot on Toys4k and ShapeNetCore.v2. Does it allow using weights pre-pre-trained on other datasets such as DINO and CLIP? Should it be fixed? There would be performance gap due to the different pre-pre-training rather than different methods. In this context, constraints should be defined clearer. E.g., the backbone should be self-supervised or the pretraining dataset should not contain the classes in the proposed dataset.
1. Number of novel categories is not defined. Is it one? Then it is too far from the useful task.
1. Hypothesis in L63 is not verified.
1. L42 A household robot example is far from the proposed task. It is mentioned in the limitation section but it is somewhat misleading in Introduction.
1. Only Categ-MOoj-SuppAssign has something to do with LSME. Results on other for data variants seem trivial. (L227 The challenge faced by the models when generalizing from instance to category level.)

**Relation To Prior Work:**

Yes. L98

**Summary And Contributions:**

This paper introduces a new task: low-shot object learning with mutual exclusivity bias (LSME).
* Object learning = associating words to objects [L27]
* Mutual exclusivity bias = word learning constraint that involves the tendency to assign one label/name, and in turn avoid assigning a second label, to a single object [wiki]
* Object learning with mutual exclusivity bias = When presented with a scene containing multiple objects, some familiar and some unfamiliar, mutual exclusivity guides the child to bind new object names to the unfamiliar objects. [L22]
  * Base classes = familiar objects are learned in advance.
  * Novel categories = unfamiliar objects
* Low-shot setup = a limited number of samples from the novel categories are available. [L106]
* Input = an RGB image with multiple object instances and a novel category name.
  * Multiple objects = 1 unknown + N-1 known categories
  * Goal = associating the novel category name and the unknown object in the image

There are new datasets for this task.

Procedure for solving this task is
1. Localizing objects in a RGB scene.
1. Classifying the objects into base / novel.
1. Assigning the novel label to the novel objects.

---

> ### Author Response · Authors · 2023-08-21
> **To Reviewer BMRQ - Answers to Opportunities For Improvement**
>
> ### Q1. “The proposed task has multiple sub-tasks that are not easily solvable given the proposed dataset.”
>
> Our experimental results demonstrate the feasibility of tackling our novel LSME task, and its associated subtasks, using the proposed dataset. There are three sub-tasks in LSME: 1) object localization, 2) open-world recognition, and 3) low-shot learning. First, for object localization, using off-the-shelf class-agnostic segmentation models (e.g. SAM) yields satisfactory performance (mIoU > 0.7). Second, during the low-shot support assignment phase where open-world recognition is required, pretrained DINOv1 and DINOv2 models give approximately 50% accuracy as compared to 33.33% for chance. Third, our experiments show that off-the-shelf self-supervised vision models yield ~57% for low-shot generalization alone while chance is 20%. Finally, the performance on LSME stands at ~40%, twice as good as chance (20%).  We note that there is significant room for improvement on all sub-tasks individually and LSME as a whole, however, the performances are far from chance. Moreover, due to the procedural nature of our data generation system, we have the ability to increase the complexity as solutions to LSME improve from the current baselines.
>
> ### Q2.1. Constraints for the baselines and benchmarking approach
>
> We thank Reviewer BMRQ for this question, which has helped us improve the presentation of our problem formulation (via the baselines and benchmarking approach.) Solving the LSME problem requires three steps: 1) Localizing the object instances within the scene via segmentation; 2) Associating the novel category label with the unknown category instance that it refers to; and 3) Solving a low-shot learning task by classifying other object instances from the novel category. The main focus of LSME is learning about novel object-category association. To prevent leakage by which pre-training data might include labeled data from the low-shot categories, we impose the following constraint: Pre-training prior to the low-shot phase cannot incorporate any object-label association data. This constraint further aligns with the investigation of a key question in developmental psychology: how can a child rapidly learn words in the absence of any prior experience with object semantics.
>
> ### Q2.2. Experimental design
> We now discuss how the constraint on pretraining affects our baseline design decisions regarding object localization, object-label assignment, and low-shot generalization.
>
> For object localization, our baseline model is built on FreeSolo. We also evaluated CutLer and SAM, SOTA class-agnostic segmentation models, in our experiments (see Table 8 in the paper). We fine-tuned a pre-trained FreeSolo using renders of 1000 ABC scenes with ground-truth instance masks. This fine-tuned model forms the foundation of our localization approach. No object labels are used at any step of this procedure, which satisfies the constraint.
>
> For object feature extraction that facilitates both object-label assignment and low-shot generalization, we focused on analyzing the performance of models initialized from DINOv1 and DINOv2 pre-trained weights as these are SOTA self-supervised vision models. Our proposed baseline for the object feature extractor is initialized with DINO weights and further fine-tuned on the ABC scene dataset before being tested for low-shot generalization (the ABC dataset does not have category labels.) Specifically, we generate scenes containing multiple ABC objects that involve occlusions. We then fine-tune DINOv1 and DINOv2 backbones using these rendered scenes. By framing the task around self-supervision we can gain insights into how LSME can be tackled without prior language inputs.
>
> We further investigated the performance of large-scale vision-language foundation models, in order to understand the difference between self-supervised and weakly-supervised pretraining approaches. We presented findings using CLIP and ImageBind pre-trained models, which violate the constraint by having unrestricted (in terms of category composition) image captions in their pre-training but have performance on LSME that is on-par with self-supervised methods without any language input (last row, table 3).
>
> ### Q2.3. “There would be performance gap due to the different pre-pre-training rather than different methods”
> We focused on initializing models with self-supervised vision pretrained weights to ensure that all methods have the same training signals available for training. In Table 5 where we compare different self-supervised finetuning methods (on Toys4K base classes and on ABC), we initialize the models with the same backbone (DINOv2 ViT-B/14). Note that we do not provide any labeled data even in the fine-tuning stage.

---

> ### Author Response · Authors · 2023-08-21
> **To Reviewer BMRQ - Answers to Opportunities For Improvement (Continued)**
>
> ### Q3. “Number of novel categories is not defined”
> In this work, we follow the standard framing of low-shot inference, such that "1-shot 5-ways” means that each episode has 5 novel categories, each with only 1 object during the low-shot training phase. However, we want to emphasize the unique differences between LSME and the standard low-shot scenario: During low-shot training, we do not provide the model with ground truth label-object associations—the model has to correctly identify the object in the scene that does not belong to any of the base categories. Furthermore, our inputs are scenes with multiple objects in that occlusions are involved, diverging from the context of only a single object being present [1,2,3]. We have clarified this in the paper.
>
> [1] Stojanov, S., Thai, A., Huang, Z., & Rehg, J. M. (2022). Learning Dense Object Descriptors from Multiple Views for Low-shot Category Generalization. Advances in Neural Information Processing Systems, 35, 12566-12580.
>
> [2] Stojanov, S., Thai, A., & Rehg, J. M. (2021). Using shape to categorize: Low-shot learning with an explicit shape bias. In Proceedings of the IEEE/CVF conference on computer vision and pattern recognition (pp. 1798-1808).
>
> [3] Padmanabhan, D. C., Gowda, S., Arani, E., & Zonooz, B. (2023). LSFSL: Leveraging Shape Information in Few-shot Learning. In Proceedings of the IEEE/CVF Conference on Computer Vision and Pattern Recognition (pp. 4970-4979).
>
> ### Q4. “Hypothesis in L63 is not verified”
> Hypothesis in L63: “We hypothesize that [the degradation of baseline performance when there are occlusions] is due to the biases induced by training on object-centric training datasets.”
>
> We empirically verify this hypothesis in section 4.3. Initially, we generate scenes containing multiple ABC objects that involve occlusions. We then fine-tune DINOv1 and DINOv2 backbones using these rendered scenes. To evaluate our models, we measure their performance on the CO3D dataset with increasingly incomplete object masks. Our model, having been exposed to occlusions in the rendered ABC scenes, shows a slower decline in performance as the amount of occlusion increases compared to other models.  Notably, even when the segmentations are masked at a 0.5 ratio (with only 50% of the object being visible), our DINOv1 ViT S/8-based model outperforms the performance of DINOv2 ViT G/14, despite having substantially fewer parameters and being trained on an order of magnitude less data. These results demonstrate the benefit of occlusion-aware feature representations.
>
> ### Q5. "L42 A household robot example is far from the proposed task. It is mentioned in the limitation section but it is somewhat misleading in Introduction."
> We agree with Reviewer BMRQ that the household robot example does not directly align with the proposed task. However, we consider this example as a motivational context for our work rather than an immediate application of the presented task. We have updated the paper and clarified this in the Introduction.
>
>
> ### Q6. “Only Categ-MObj-SuppAssign has something to do with LSME. Results on other for data variants seem trivial. (L227 The challenge faced by the models when generalizing from instance to category level.)”
> We would like to emphasize that only the first data variant (Inst-SObject) is relatively straightforward as it does not require category generalization. The subsequent variants (Categ-SObject, Categ-SOject-PoseVar, and Categ-MObject) require instance-to-category generalization, as in the standard low-shot learning setting. Although not requiring mutual exclusivity bias, these variants are significantly more challenging by introducing pose variability and multiple objects.
>
> The goal of these experiments is to highlight multiple factors that make LSME a challenging task and the capabilities that models need to have to succeed at LSME. Further, these data variants can be used to diagnose the limitations of models during development. For example, our results show that current SOTA methods experience difficulty when occlusions are introduced, while they appear to be less sensitive to various pose changes. We noticed that the heading of Section 4.2 can be confusing since some of the data variants do not require mutual exclusivity bias. We have changed this heading in the draft.

---

> > ### Comment · Reviewer_BMRQ · 2023-08-28
> >
> > I have read the rebuttal and it resolves my concerns. I updated my score from 5 to 6.
> >
> > Please check other sections too: correctness, clarity, and typos.

---

### Author Response · Authors · 2023-08-21
**To All Reviewers**

We thank all the reviewers for their insightful feedback. We are pleased to find that reviewers found our work "novel" (Reviewer HVNJ), "interesting" (Reviewer xYTB, Reviewer FRCv) and "meaningful" (Reviewer BMRQ). We addressed each reviewer's concern individually below as well as updated our paper. Changes made in the updated paper are highlighted in blue.

**Contribution and Novelty:** Our contributions in this work are 4-fold:
1. We are the first to provide a computational framing of mutual exclusivity bias via LSME task.
2. We provide a procedural data generation pipeline that enables the creation of progressively more challenging dataset for LSME using any 3D asset.
3. We demonstrate an extensive study for benchmarking the performance of various methods including foundation models on LSME and related tasks.
4. We introduce a novel baseline method that defines SOTA on LSME.

---

### Decision · Program_Chairs · 2023-09-22

**Decision:**

Accept (Poster)

**Comment:**

In this paper, the author proposes a new benchmark method and evaluates its performance. The Performance benchmark testing is performed on multiple baseline models, including methods based on the basic model.  And a new benchmark method is proposed to achieved state-of-the-art results in LSME tasks. Specifically, the author evaluates the performance of the benchmark method by measuring metrics such as support allocation accuracy (SA), low sample accuracy (LSA), and mIoU for instance segmentation.